# The Effectiveness of Robot-Enacted Messages to Reduce the Consumption of High-Sugar Energy Drinks

Isha Kharub [1,*], Michael Lwin [1], Aila Khan [1], Omar Mubin [2] and Suleman Shahid [3]

[1]  School of Business, Western Sydney University, Penrith 2751, Australia; m.lwin@westernsydney.edu.au (M.L.);
    a.khan@westernsydney.edu.au (A.K.)
[2]  School of Computer, Data and Mathematical Sciences, Western Sydney University, Penrith 2751, Australia;
    o.mubin@westernsydney.edu.au
[3]  Computer Human Interaction and Social Experience Lab (CHISEL), Syed Babar Ali School of Science and
    Engineering (SBASSE), Lahore University of Management Sciences, Lahore 54972, Pakistan;
    suleman.shahid@lums.edu.pk
*   Correspondence: i.kharub@westernsydney.edu.au

**Abstract:** This exploratory study examines the effectiveness of social robots' ability to deliver advertising messages using different "appeals" in a business environment. Specifically, it explores the use of three types of message appeals in a human-robot interaction scenario: guilt, humour and non-emotional. The study extends past research in advertising by exploring whether messages communicated by social robots can impact consumers' behaviour. Using an experimental research design, the emotional-themed messages focus on the health-related properties of two fictitious energy drink brands. The findings show mixed results for humour and guilt messages. When the robot delivered a promotion message using humour, participants perceived it as being less manipulative. Participants who were exposed to humourous messages also demonstrated a significantly greater intent for future purchase decisions. However, guilt messages were more likely to persuade consumers to change their brand selection. This study contributes to the literature as it provides empirical evidence on the social robots' ability to deliver different advertising messages. It has practical implications for businesses as a growing number seek to employ humanoids to promote their services.

**Keywords:** consumer research; message appeals; humour; guilt; humanoid social robot; advertising; experimental design—health prevention





## 1. Introduction

Social robots provide a unique opportunity for businesses to communicate with existing and potential consumers. Traditionally, health-related messages have been relayed via personal channels of communication (e.g., sales staff), as well as a range of different media vehicles (e.g., television, radio, outdoors, etc.). It is well-established that most traditional media vehicles such as television and radio are designed for a one-way mode of communication, and do not engage with the audience. In fact, audiences are passive recipients of such advertising messages [1,2]. However, social robots are more engaging and can deliver individualized messages to the consumer. These communications are important in raising awareness of health-related issues that are connected to high-sugar beverages. However, it is unknown how the messages delivered by a social robot will be perceived by young consumers.

Research suggests that in a retail setting, salespeople can significantly impact consumers' perceptions [3]. The face-to-face communication is recognized as possessing the capacity to build relational meaning in communications [4]. Despite the demonstrated importance of the salesperson, studies show that consumers find retail services to be mediocre and in decline [5]. A number of factors could be linked to service failures, including the

salespersons' service behaviours [6]. In a service setting, it is not uncommon to find employees experiencing emotional exhaustion [7] and, unintentionally, showing their stress levels when communicating with customers. Therefore, scholars have suggested the need to identify low cost, feasible alternatives which assist employees, while also enhancing consumer perceptions [8].

The importance and speculated benefits of social robots over more traditional forms of interaction have been a contentious issue in Human–Robot Interaction [9]. Robots or social robots, per se, are accepted to have a wide range of output modalities that can enhance their interaction with humans. By leveraging their physical embodiment and appearance, they can readily enrich the interaction with a user [10,11]. Social cues and emotions exhibited through their gestures, facial expressions, emotions, and voice pitch provide a natural experience to the interacting user [12]. Such tools are of great importance in a typical retail scenario as businesses look to integrate robots in retail settings, e.g., [13–18]. However, the use of social robots as an advertising or a communication agent has been largely ignored by researchers, e.g., [19]. Most of the studies in this area are limited to how social robots can influence customer experience in a service domain.

In public health advertising, emotional appeals are more effective at influencing consumers' decisions [20]. In a human–human interaction, it is easier to transmit emotional cues (e.g., crying to show sadness). However, this task is more challenging for social robots due to technological limitations. Therefore, would it be possible for social robots to deliver a public service announcement using emotional messages to influence consumers? Previous studies in this area of research have explored the verbal and non-verbal cues of a robot to deliver a message [21]. However, there is a lack of empirical research on the effectiveness of advertising appeals when used in a human–robot interaction scenario, particularly in a health prevention context. The research gap in the literature highlights the need to understand the effectiveness of emotional advertising appeals delivered by robots in a business setting. This exploratory study aims to address these research gaps by exploring how a social robot can effectively evoke consumer responses (via guilt and humour messages) and, ultimately, persuade consumers to change their purchase decisions.

## 2. Relevant Literature

### 2.1. Social Robots

Humanoid robots are a form of social robots, and they can exhibit social behaviours and create human-like interactions. This new breed of robots can identify and comprehend human emotions, respond to commands, and function adequately in a social context. Social robots have the ability to detect human emotions [22,23] and respond in multiple languages [24]. They learn from previous interactions with humans and each other, and they have been described as employees of the future for many industries [25]. For example, over 2000 companies are using Pepper (a humanoid social robot), to greet customers and provide directions and information to visitors (i.e., customer experience and interaction) [26].

The human-like appearance of humanoid robots tends to induce positive perceptions and attitudes amongst customers and increases emotional attachment [27–29]. Research has shown that customers place high importance on their relations with the service employees such as rapport, engagement, and trust, and thus, providing social and emotional value [25]. Therefore, the acceptance of a social robot in society is dependent not just on the fulfillment of functional needs but also on socio-emotional and relational needs [25]. Thus, humanoid social robots may prove to be ideal candidates as front-end sales staff [11].

Due to the popularity of social robots for businesses, researchers have attempted to understand how the technology could be applied in the real world. The research in human-robot interaction (HRI) accounted for 58.8% of all articles in social robotics [30,31] and highlights the importance of this area of research. Increasing media attention towards social robotics, advancement in the robot's ability to perform in unstructured environments has driven studies to examine the acceptance of robots in society and has driven innovation in science and technology [32,33]. Scholars have used field trials in shopping malls to test

the robot's influence on people's daily life, e.g., [34]. Studies show that the number of people that were exposed to the advertisement was higher when a robot was present [35]. Furthermore, other studies have explored the robot delivering the advertising message using greetings and dancing, e.g., [36]. However, the delivery of the message appeal by the robot has been largely ignored and it highlights a key gap in the literature.

To assist with the adoption of social robots by businesses, it is vital to understand the role of persuasion [37]. This is a critical factor for advertising messages. A robot should be persuasive and be able to change or influence the decision of a consumer. A wide range of scenarios have been studied pertaining to studying persuasive social robots [38], such as energy consumption [39], education [40], or service [40], but application in health prevention is still limited.

*2.2. Message Appeals*

In the field of marketing communications, an advertising appeal "is the basic idea that the advertiser wants to communicate to the audience . . . " [41–46]. It refers to the theme of the message, which assists in communicating the message content [42]. An emotional appeal consists of positive (e.g., humour) or negative (e.g., guilt) emotional elements (see Figure 1) [43]. Examples of emotional appeals include fear, guilt, shame, and humour [44]. By using the right appeal, marketers consciously attempt to motivate potential consumers toward a behaviour (e.g., purchasing a product) or to influence them to change their attitude or conception of an advertised product [45].

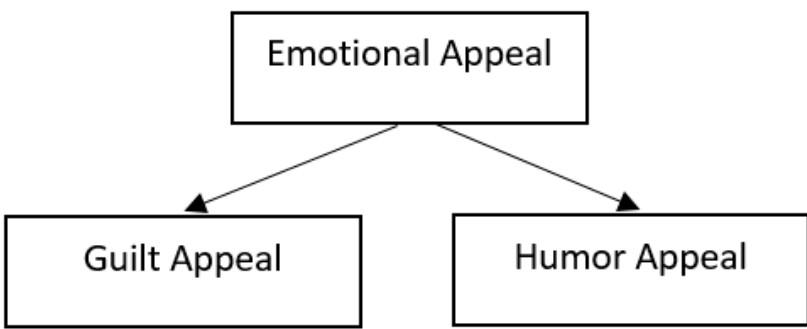

**Figure 1.** Types of emotional appeal.

Guilt Appeal vs. Humour Appeal vs. Non-Emotional Appeal

Emotional messages are powerful for advertisers, as these can affect consumers' attention [46,47], the audience's reaction to advertisements, e.g., [47], and influence brand attitudes, e.g., [48]. This study explores the effect of two types of emotional appeals: guilt and humour appeal and non-emotional appeal.

*Guilt* is an action-oriented emotion, which means that when a person feels guilty, they will act to remove the feeling [49]. The Negative State Model explains that individuals will seek to reduce negative emotions such as guilt [50]. The feeling of guilt is aroused when there is a comparison between one's actual and one's optimal behaviour, according to the socially defined standard, rule, or moral imperative [49,51]. For example, as a society, we believe that we should eat healthily. When we consume sugary products, we are going against socially held beliefs and standards. This arouses a feeling of guilt. Advertisers trigger guilt through their messages and then provide a solution on how consumers can reduce the feeling of guilt (e.g., purchase the product/service) [52]. A number of studies have demonstrated the effectiveness of guilt appeals in the form of attitude change or increased behavioural intentions [52–55].

Advertisers have used a number of methods to evoke guilt: statements of facts, suggestions, and questions are commonly used techniques [56]. However, in the case of social robots, guilt may also be evoked using non-verbal cues such as facial expression, body language, and hand movement, e.g., [21]. Studies in other areas of research suggest animated voice agents could have a direct impact on how consumers absorb the information [57].

However, these studies are limited to consumer behaviour in a human–human interaction, e.g., [58]. Thus, research is needed to understand how social robots can use emotions effectively.

Guilt appeals in health-related messages are used to evoke feelings of remorse and responsibility, which in turn mobilizes individuals to take remedial action [59]. It has been argued that guilt as an emotion can be channelled in constructive ways to motivate individuals to follow a course of action [56]. Guilt appeal tends to create a negative mood and encourages the viewer to take action to remove the feeling of guilt [58,60]. Previous health research has shown that guilt can encourage the avoidance of risky behaviours such as binge drinking [61]. Some studies have shown that guilt appeals can reduce the consumption of sugary beverages, e.g., [62]. Therefore, it is predicted that guilt appeal will have a significant impact on changing behaviour. However, social robot studies that use guilt messages to reduce this behaviour are largely ignored by scholars.

*Humour* in an advertising appeal is determined on the basis of the use of puns, jokes, understatements, turns of phrases, double entendre, satire, irony, slapstick, or incongruity [63]. Humour is also one of the most commonly used appeals in advertising and is effective in shaping attention and liking towards the advertisement [64]. Messages with a humour-appeal attempt to make people laugh and create a positive mood [63].

Humour appeals in health-related messages create a positive affective response by evoking laughter and, in turn, create a positive attitude towards the advertisement. Humourous messages also have a positive effect on information processing, e.g., [65] as an individual is in a positive mood. Thus, the individual's tendency to counterargue against the persuasive message is reduced, e.g., [66]. Recent research with social robots has shown that both verbal and non-verbal behaviours are equally important in delivering a humourous message [67]. When humour is used by a robot, it increases the likability of the robot [68,69] and has a positive impact on perceived enjoyment in undertaking a task [12,69].

*Non-emotional* messages use factual information or generic statements to persuade the customer. An advertisement is informative if the consumer perceives the information as important and relevant [70]. Non-emotional messages use facts, features, or product attributes to showcase the quality of the product. The literature suggests the advertisement's informative factor is a good predictor of advertisement likeability and brand attitude [71]. Factual information about a brand may create a better understanding of the brand and consequently helps consumers during the decision-making process [72]. The key facts are often used in non-emotional health advertisements for reducing heart rate, blood pressure, weight gain or loss, vitamin content, and exercise routine, e.g., [73].

### 2.3. Key Advertising Metrics

In this study, a social robot delivers health-oriented messages using guilt, humour, and a non-emotional appeal. The underlying objective of the message is to communicate the health-related attributes of the products. Advertisers use *message credibility, inferences of manipulative intent, attitude and purchase intention and changes in brand selection* to evaluate the effectiveness of the message appeals [74,75]. These key variables provide advertisers to compare the effectiveness between different message executions.

*Message credibility* is defined as the extent to which the consumer perceives claims made about a product/brand are truthful and believable [76]. Cognitive Response Theory implies that when consumers perceive communications or arguments about the brand as credible, their cognitive responses and attitude towards the message will be more positive [77]. The theory implies that when consumers perceive the message to be credible, they are more likely to believe in the message and purchase the product [58]. Therefore, the message delivered by a social robot needs to be perceived as credible by the consumer. It is predicted that non-emotional messages are perceived to be more credible than emotional messages, as they contain more factual information about the product.

**Hypothesis 1 (H1).** *There will be significant differences in participants' rating for credibility for each type of advertising message.*

*Inferences of manipulative intent* are defined as consumers' opinions as to whether the message-sender (i.e., the robot, in this study) is attempting to persuade people by using inappropriate, unfair, or manipulative means [78]. Attention-grabbing techniques could lead consumers to perceive that the marketer is attempting to manipulate or unfairly persuade them. If a consumer believes that the robot is using manipulative tactics, then it negatively impacts attitudes towards the message and the product [79]. Humour messages are perceived to be more entertaining than guilt and non-emotional messages. Therefore, individuals are more likely to view humour messages as an appropriate means of persuasion.

**Hypothesis 2 (H2).** *There will be significant differences in participants' rating for inferences of manipulative intent (of the robot) for each type of advertising message.*

*Attitude* is defined as, "the predisposition to respond in a favourable or unfavourable manner to a particular message stimulus . . . " [76]. Past research has shown that consumers' favourable attitudes towards the message lead to a greater likelihood of purchase of the product [80–82]. As mentioned previously, guilt messages tend to create a negative mood, while humour messages create a positive mood. In comparison, the non-emotional messages create a more neutral mood. Thus, the following is hypothesised.

**Hypothesis 3 (H3).** *There will be significant differences in participants' rating of attitude (towards the robot) for each type of advertising message.*

*Purchase intention* is defined as the buyer's willingness, the probability, and the possibility to purchase a product or service [83]. Intention is considered to have a high impact on behaviour [84]. Past researchers have shown that advertising messages can positively impact consumers' purchase intentions [85–88]. Guilt is an action-oriented emotion; thus, it is more likely to evoke a behaviour than humour. Researchers have suggested emotional messages are more effective than non-emotional appeals [89]. Thus, the following is predicted.

**Hypothesis 4 (H4).** *There will be significant differences in participants' rating of purchase intentions for each type of advertising message.*

*Changes in Brand Selection* [90] has been conceptualised as the respondent's brand switching behaviour, which is expected to take place after exposure to product-related information delivered by the robot. This is a common strategy used by advertisers to promote healthier alternatives to the sugar drinks. For example, the advertisement for the Sugar-Free or Diet Coke highlights lower sugar content to suggest it is healthier than the regular Coke. Therefore, as a preventative measure, should social robots be positioned next to vending machines to highlight the consequences of consuming sugary drinks? It is predicted that guilt is more likely to evoke a change in the brand selection due to a violation of one's standards. Therefore, the individual is more likely to conform to the guilt message than the humour message.

**Hypothesis 5 (H5).** *There will be significant differences in participants' brand selection for each type of advertising message.*

### 3. Methodology

To conduct the research a $1 \times 3$ (Guilt Appeal vs. Humour Appeal vs. Non-emotional appeal) between-subjects experiment was undertaken during November 2019—February 2020 at a very large university in Sydney, NSW. The study adapted and adopted the methodology by (60) to a social robot context. The experimental protocol and procedure were approved by the university's ethical review board.

The respondents interacted with the robot before completing the survey. The social robot known as Nao was used for the study. Nao is developed by Softbank Robotics, and it is one of the most popular small humanoid robots on the market (Softbank Robotics). The robot has speech and voice recognition, touch sensors, and 25 degrees of freedom for expression and movement.

The sample consisted of university undergraduate students from computing and business. A pool of students was randomly selected to complete the survey. Younger consumers are more likely to engage with the social robots [91] and thus were selected for the study. Further, the respondents had similar "life stages" and therefore their decisions were less influenced by external factors.

Respondents were given a scenario and were asked to imagine that "It is a hot afternoon and after a long day at the university, you need a pick-me-up drink before the next class. As you are walking, you see two new energy drinks—ENERGISE and Hyped-Up—in the vending machine. You have to choose one of these drinks." After the participants have selected one of the drinks, the social robot that is standing next to the vending machine pops up and delivers a message (a prevention strategy to reduce the consumption of high-sugar energy drinks) (See Figures 2–4).

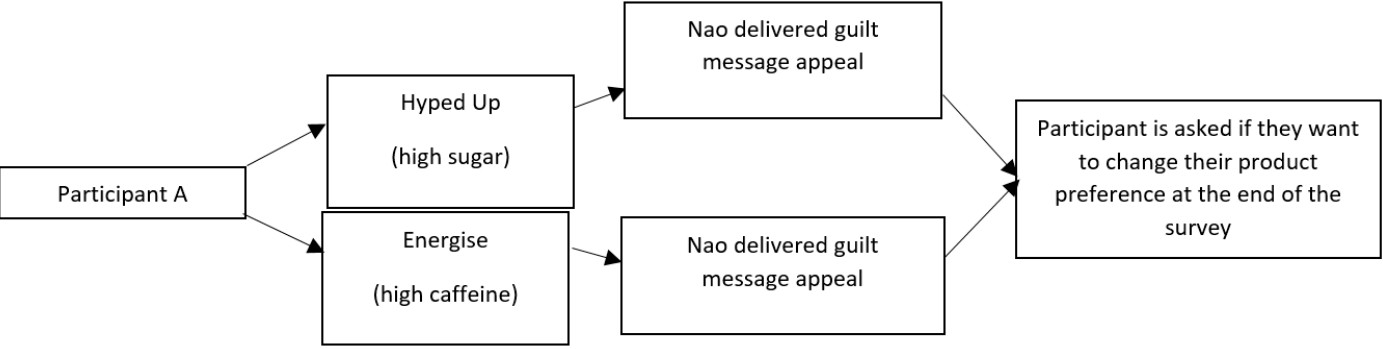

**Figure 2.** The guilt appeal.

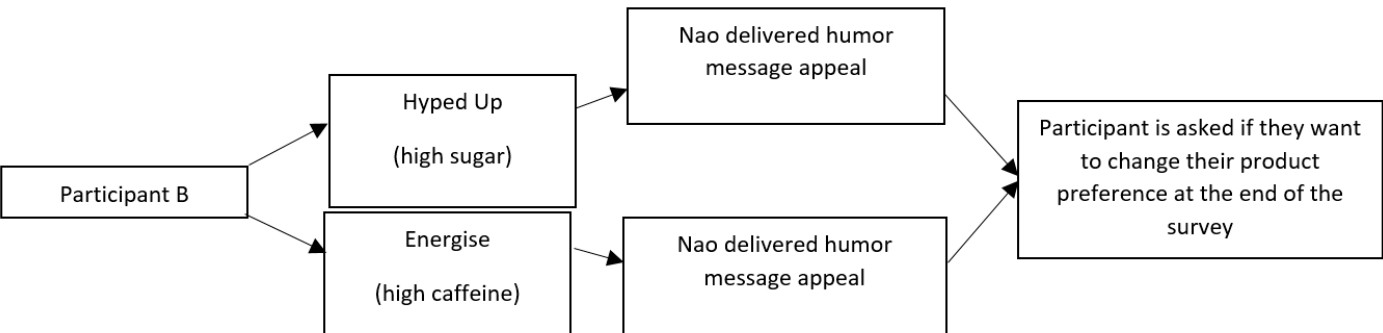

**Figure 3.** The humour appeal.

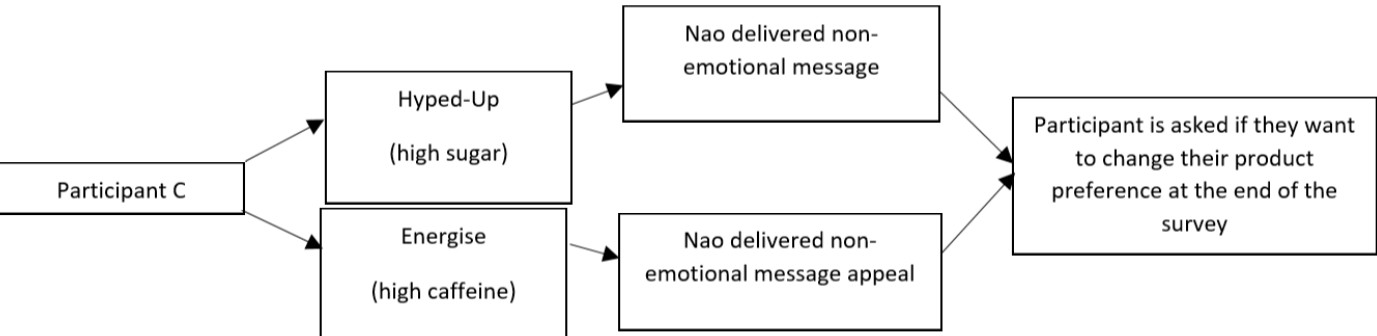

**Figure 4.** The non-emotional appeal.

The total number of participants was 206. Participants were undergraduate university students, with more than 90% in the age group 18–24 years (see Table 1). This group was selected as they are more likely to engage with social robots than other groups [91]. To compare the differences between the experiments, a homogeneous sample is required to control for external factors [92]. By limiting participants to the same "life stage" (in this case students) the researcher can control and reduce the external factors that may influence the respondent's proneness to guilt [93]. None of the participants received academic credit for their participation in the study. However, the research activity was connected to a report. Students were required to develop a report (worth 10% of the unit) that analyzed the effectiveness of the message. The report asked the following questions: (1) Is the message effective? (2) What advertising appeal was used? (3) How does the message change consumers' decision. Previous studies that used student samples with a course credit reported no significant biasing to the result [94,95]. In fact, the students were highly engaged and motivated to participate in the research. Overall, there was a slightly higher number of male participants ($n = 110$; 56.5%) than female participants ($n = 96$; 43.5%) (see Table 1). In line with the university's overall student population, 55% of the participants came from a non-English speaking background. However, all students spoke fluent English.

**Table 1.** The respondent profile.

| Age | Gender | Education |
|---|---|---|
| 18–24—90% | Males—55% Females—45% | Undergraduate Business students—100% |

Almost an equal number of participants were assigned to each of the two experimental groups, depending on their preference for fictitious energy drinks: "ENERGISE", which is high in caffeine ($n = 69$) and "Hyped-Up", with a higher quantity of sugar ($n = 68$). Participants were separately recruited for the control group ($n = 69$). Care was taken to ensure that participants could not view/hear messages of other treatment groups.

*3.1. Stimuli*

Sugary drinks and diet drinks are not considered an essential part of a healthy diet and only a limited intake of these drinks is recommended as they have very little nutritional value. A high intake of these drinks can lead to adverse health conditions such as Type 2 diabetes, cardiovascular diseases, high blood pressure, obesity, and an increased risk of weight gain [96].

Young adults consume the highest number of sugary drinks with 61.3% consuming them at least once a week [96]. In Western Sydney, 56% of teenagers drank fruit juice daily, whereas 13% drank energy drinks daily [97]. The consumption of energy drinks in young adults can lead to headaches, sleeping difficulties, and heart palpitations, primarily due to the high amount of caffeine contained in them [98].

*Brand Selection.* Two fictitious energy drinks were used to test the hypothesis. Fictitious products have been used in previous studies to measure attitudes, perception, and behavioural intentions towards an ad and towards the brands [99,100]. Previous studies show that the results are valid and appropriate because it avoids any prior attitude towards the brands [101]. The two products are Hyped-Up and ENERGISE (see below for product details). The participants were asked to pick one of these.

Types of Message Appeals

Two types of emotional appeals (guilt vs. humour) were designed as the experimental conditions for each energy drink. A non-emotional appeal was created as the control condition. Participants were equally assigned to each of the three conditions.

Prior to running the experiment, to ensure that the messages were relevant and evoked guilt and humour, the message stimuli were tested with 35 undergraduate business students. The participants were shown the three messages one at a time (guilt, humour, and non-emotional) and then they were asked to rate whether the message evoked guilt, humour, or non-emotional on a seven-item Likert scale. The test shows that the guilt message evoked guilt emotions (Guilt mean = 4.06), the humour message evoked humourous reactions (Humour mean = 4.08), and the non-emotional message evoked a rational response (Non-emotional mean = 4.89).

The results also show low awareness of the two energy drinks. The results suggest these two brands were appropriate as participants had no prior knowledge of or attitude towards the brands. Thus, their prior experience with the brands will not influence their decision during the study.

The participants were also asked about their attitude towards the two brand names to ensure that the brands were perceived positively. The four-item measure for the brand name is: "positive", "good", "interesting", and "attractive". The items were measured on a seven-item Likert scale. Participants perceived both brands positively; ENERGISE brand name (mean = 5.00) and HypedUp brand name (mean = 4.50).

As given in Table 2 guilt messages used statements to spell out the harmful effects of consumption, whereas humourous messages used a joke to communicate the main idea.

**Table 2.** Types of message appeals.

| | |
|---|---|
| Guilt Messages | **ENERGISE (High Caffeine):**<br>*"Studies show high level of caffeine intake can lead to anxiety and higher blood pressure which can cause stroke"* |
| | **Hyped-Up (High Sugar):**<br>*"Studies show high level of sugar intake can lead to type 2 diabetes and weight gain"* |
| Humour Messages | **ENERGISE (High Caffeine):**<br>*"What do you call a robot high on caffeine? The Caffeinator . . . hahahahaha"* |
| | **Hyped-Up (High Sugar):**<br>*"What do you call a band with high levels of sugar? The Diabeatles . . . hahahahaha"* (a play on word with diabetic and The Beatles) |
| Non-Emotional Messages (for Control Group) | **ENERGISE Brand (High Caffeine):**<br>*"ENERGISE has only 10 g of caffeine and 50 g of sugar"* |
| | **Hyped-Up Brand (High Sugar):**<br>*"Hyped-Up has only 10 g of sugar and 50 g of caffeine"* |

All the message appeals were followed by a suggestion to choose a different brand. Messages for the control group did not carry any emotional appeal. As shown in Table 2, control group messages carried factual information only.

*3.2. Procedure*

The experiment was conducted with one participant at a time. Participants were recruited in the main lobby of the university's campus. All participants were required to sign consent forms and read the study's information sheet, as required by the university's Ethics Committee. Participants were then escorted to a study room where the experiment was to be conducted, in a lab setting. The room was equipped with a computer to collect survey responses.

**Activity 1**: As the first step of running the experiment, participants were presented with printed names and descriptors of the two hypothetical brands (see Table 3). Sufficient time was allocated for participants to read the brand's nutritional composition.

**Table 3.** Fictitious energy brands and their composition.

| | |
|---|---|
| **ENERGISE** | 10 g of sugar and 50 g of caffeine. One drink can increase your attention and reaction speed by 20% |
| **HYPED-UP** | 50 g of sugar and 10 g of caffeine. One drink can increase your attention and reaction speed by 20% |

**Activity 2**: Participants were then presented with a written scenario which set the scene (see above for the scenario). In view of the presented scenario, participants were asked to choose between the two energy-drink brands. A Nao robot was present in the room throughout.

**Activity 3**: Once the participant had selected the brand, the robot delivered one of the three messages (guilt, humour, or non-emotional). The robot's dialogues were handled by a researcher through a Wizard of Oz setup.

**Activity 4**: Immediately after receiving the brand-related message from the robot, participants were asked to rate their response to the message on the following variables: *message credibility*, *inferences of manipulative intent*, *attitude*, and *purchase intention*. Responses were recorded via the Qualtrics online survey system. For most participants, it was their first interaction with a social robot.

**Activity 5:** Finally, participants were asked if they would like to change their product preference after the robot delivered the message. This response was used to calculate any change in consumers' brand preferences. The survey then concluded with general demographic information (age, gender, income). The survey structure is shown in Table 4. For example, participant A initially selected Hyped-Up. The robot will deliver the guilt message "Studies show high level of sugar intake can lead to type 2 diabetes and weight gain. Would you like to change your choice to ENERGISE? It has less sugar than Hyped-Up."

**Table 4.** The survey structure.

| |
|---|
| Information Sheet and consent |
| Scenario (Hot afternoon) |
| Initial Brand Selection (ENERGISE or Hyped-Up) |
| Message (Humour, Guilt, Non-emotional) |
| Consumer response (Measures: Message Credibility, Inferences of Manipulative Intent, Attitude, Purchase Intention) |
| Post-Message Change in Brand Selection (ENERGISE or Hyped-Up) |
| Demographics (Age, gender, income) |

*3.3. Measurements*

A seven-point Likert scale was used to measure the most dependent variables. All the scales were adapted from established sources: three items for message Credibility [60], six items for Inferences of Manipulative Intent [78], three items for Attitude [76], and three

items for Purchase Intention [87]. Exploratory Factor Analysis and scale reliability were tested separately for the three messages (humour, guilt, control). Inferences of Manipulative Intent were re-coded to a positive valence (e.g., "I think that this robot provided a fair perspective"). A high Inference of Manipulative Intent mean score indicates a low level of manipulative intent from the robot. A summary of the items and their associated reliability scores (Cronbach Alpha) is provided in the table (see Table 5).

**Table 5.** Measurements: Reliability of Survey Items.

| Scale | Scale Reliability ($\alpha$) |
|---|---|
| **Message credibility** [56]<br>This robot's message is believable<br>This robot's message is truthful<br>This robot's message is realistic | 0.772–0.89 |
| **Inferences of manipulative intent** [71]<br>The way this robot tries to persuade people seems acceptable to me<br>The robot tried to manipulate the audience in ways I do not like<br>I was annoyed by this robot because it was trying to inappropriately manage or control the audience<br>The robot tried to be persuasive without being excessively manipulative<br>The robot was fair in what was said and shown<br>I think that this robot provided an unfair perspective (reverse coded) | 0.610–0.720 |
| **Attitude towards the robot** [69]<br>"I have good attitude towards the robot"<br>"My attitude towards the robot is favourable"<br>"My attitude towards the robot is positive"<br>"I dislike this robot" | 0.780–0.810 |
| **Purchase intention** [80]<br>"It is very likely that I will buy this product"<br>"I will purchase this product the next time I need it"<br>"I will definitely try this product" | 0.860–0.890 |

## 4. Data Analysis and Results

Initially, the researchers analysed descriptive statistics for the three groups. Table 6 provides the averages scores for all measures across the two experimental groups and the control group. As can be seen in the table, amongst the listed variables, "*attitudes towards the robots*" received the highest ratings from participants. This shows that people's attitudes toward social robots remains strong no matter which type of message appeal is used. Among the different types of appeals, messages using a humour appeal received the highest scores. This demonstrates participants' overall positive evaluations of a humourous message delivered by a robot.

**Table 6.** Average scores for the experimental groups and the control group.

| | Guilt Mean | Guilt Std. Deviation | Humour Mean | Humour Std. Deviation | Control Group Mean (Non-emotional) | Control Group Std. Deviation (Non-Emotional) |
|---|---|---|---|---|---|---|
| Message Credibility (H1) | 4.93 | 0.915 | 4.94 | 1.08 | 4.86 | 1.25 |
| Manipulative Intent (recoded to manipulation is acceptable) (H2) | 4.67 | 0.952 | 5.05 | 0.906 | 4.98 | 0.754 |
| Attitude towards the robot (H3) | 5.16 | 0.975 | 5.45 | 1.04 | 5.30 | 1.01 |
| Purchase Intentions (H4) | 3.72 | 1.35 | 4.48 | 1.46 | 4.13 | 1.41 |

Analysis of Variance (ANOVA) was used to test the hypotheses between subjects (see Table 7), and the results revealed that message appeal had a significant effect on IMI (F = 3.67, *p* = 0.03 and PI (F = 4.92, *p* = 0.008). A Bonferroni Post Hoc also showed that

Humour was rated significantly more positive than Guilt on IMI ($p$ = 0.034) and on PI ($p$ = 0.006).

**Table 7.** The test of between subject effects.

| Dependent Variable | Types of III Sum of Squares | df | Mean Square | F | Sig |
|---|---|---|---|---|---|
| AdCr | 0.309 | 2 | 0.154 | 0.129 | 0.879 |
| IMI | 5.611 | 2 | 2.805 | 3.665 | 0.027 |
| AdRo | 2.937 | 2 | 1.468 | 1.437 | 0.240 |
| PI | 19.616 | 2 | 9.808 | 4.920 | 0.008 |

Post Hoc Test results clearly indicated that the guilt-appeal message was not as effective as a humour appeal message in triggering low inferences of manipulative intent and purchase intentions amongst participants' (Table 8). A high inference of manipulative intent mean score indicates a low level of manipulative intent from the robot (i.e., using the 7-point Likert Scale average score, 1 = high inference of manipulative intent and 7 = low inference of manipulative intent). Table 6 explains that Guilt manipulative intent mean was 4.67 and Humour was 5.05. Therefore, the mean difference between Guilt and Humour was −0.38. The results indicate that guilt had a higher inference of manipulative intent than humour appeal (Mean Difference = −0.3824, $p$ = 0.034).

**Table 8.** The post hoc tests.

| Dependent Variable | (I) Control, Guilt, Humour | (J) Control, Guilt, Humour | Mean Difference (I − J) | Std Error | Sig |
|---|---|---|---|---|---|
| AdCr | Control | Guilt | −0.0773 | 0.18657 | 1.000 |
| | | Humour | −0.0861 | 0.18725 | 1.000 |
| | Guilt | Control | 0.0773 | 0.18657 | 1.000 |
| | | Humour | −0.0088 | 0.18725 | 1.000 |
| | Humour | Control | 0.0861 | 0.18725 | 1.000 |
| | | Guilt | 0.0088 | 0.18725 | 1.000 |
| IMI | Control | Guilt | 0.3043 | 0.14895 | 0.127 |
| | | Humour | −0.0781 | 0.14949 | 1.000 |
| | Guilt | Control | −0.3043 | 0.14895 | 0.127 |
| | | Humour | −0.3824 * | 0.14949 | 0.034 |
| | Humour | Control | 0.0781 | 0.14949 | 1.000 |
| | | Guilt | 0.3824 * | 0.14949 | 0.034 |
| AdRo | Control | Guilt | 0.1413 | 0.17207 | 1.000 |
| | | Humour | 0.1515 | 0.17270 | 1.000 |
| | Guilt | Control | −0.1413 | 0.17207 | 1.000 |
| | | Humour | −0.2928 | 0.17270 | 0.275 |
| | Humour | Control | 0.1515 | 0.17270 | 1.000 |
| | | Guilt | 0.2928 | 0.17270 | 0.275 |
| PI | Control | Guilt | 0.4106 | 0.24039 | 0.267 |
| | | Humour | −0.3451 | 0.24127 | 0.463 |
| | Guilt | Control | −0.4106 | 0.24039 | 0.267 |
| | | Humour | −0.7557 * | 0.24127 | 0.006 |
| | Humour | Control | 0.3451 | 0.24127 | 0.463 |
| | | Guilt | 0.7557 * | 0.24127 | 0.006 |

* Significance value <0.05.

For purchase intention, a high purchase intention mean score indicates a higher likeability of a purchase (i.e., 1 = low purchase intention and 7 = high purchase intention). Table 6 explains that Guilt purchase intention mean was 3.72 and Humour was 4.48. Therefore, the mean difference between Guilt and Humour was −0.76. It was revealed that messages with a guilt appeal resulted in significantly lower scores for purchase intentions than humour appeal (Mean difference = −0.7557 $p$ = 0.006). Yet, a greater percentage of participants exposed to a guilt message voluntarily changed their brand selection. This indicates the utility of using a guilt appeal when aiming to bring about a change in behaviour among a population group. This is in line with what is suggested in the literature [60,102]. Guilt is an action-oriented emotion, and when someone feels guilt, the individual takes action to remove the feeling of guilt. Therefore, the individual is more likely to change the brand choice as a result of feeling guilt. However, the results confirm that guilt messages delivered by the social robot at the point of purchase do not work. Humour was shown to be more effective at influencing consumers' behaviour. Due to greater than two factors, we performed pairwise post hoc comparisons on the individual measurements. Empirical research has identified post hoc test as a suitable methodology to test for muti-factor hypothesis [103].

Finally, this study also measured changes to participants' brand selection (Table 9). As mentioned above, participants were asked to choose between two brands (see Procedure, Section 3.2) before being exposed to the message by the robot. After receiving the message, the survey also measured whether participants were interested in changing their selection. Around twenty-five percent of consumers who received the guilt-appeal message changed their product selection.

**Table 9.** Change in brand selection and the comparison between groups.

| Brand Selection | Humour Appeal | Guilt Appeal | Control |
|---|---|---|---|
| Changed from initial decision | 20.6% (14) | 24.60% (17) | 17.40% (12) |
| Did not change from initial decision | 79.40% (54) | 75.35% (52) | 82.60% (57) |
| **Total** | 100% (68) | 100% (69) | 100% (69) |

A Chi-Square test (see Table 10) was also conducted to verify the statistical significance of the differences. No significance was reported in all three appeals ($p$ = 0.576). This is an unexpected result, and it adds to the literature and suggests that the message type does not influence behaviour change when the message is delivered by a social robot at the point of purchase. This suggests that the change in the brand selection is not driven by the message appeal. Therefore, all of these messages perform equally as a preventative strategy to reduce the consumption of high-sugar energy drinks.

**Table 10.** Chi-Square tests.

| | Value | df | Asymptotic Significance (2-Sided) |
|---|---|---|---|
| Pearson Chi-Square | 1.102 [a] | 2 | 0.576 |
| Likelihood Ratio | 1.101 | 2 | 0.577 |
| Linear-by-Linear Association | 1.092 | 1 | 0.296 |
| **N of Valid Cases** | **206** | | |

Significance value (a) = 0.05.

## 5. Discussion and Conclusions

This research aimed to test the effectiveness of two different types of emotional appeals in messages delivered by a robot in a marketing context. A summary of hypotheses and decisions are presented in Table 11. Our results indicate that while humour appeal messages

received the highest scores on all measures, there are no significant differences to non-emotional messages. Guilt appeal messages generated significantly higher feelings of the robot's manipulative intent in comparison to the non-emotional appeal message. Not surprisingly, participants' purchase intentions were significantly low for the guilt appeal message as well. These results clearly highlight that robot-delivered guilt-appeal messages are not positively evaluated by participants. However, when measuring participants' change in brand selection, the results show that guilt appeal worked the best. The largest number of participants changed their initial brand selection after being exposed to a guilt message.

**Table 11.** A summary of the hypotheses and decision.

| Hypothesis | Proposed Statement | Decision |
|:---:|:---:|:---:|
| H1 | Significant differences in participants' rating for *credibility* for each type of advertising message | Not Supported |
| H2 | Significant differences in participants' *inferences of manipulative intent* (of the robot) for each type of advertising message | Partially supported for Guilt Appeal |
| H3 | Significant differences in participants' rating of *attitude* (towards the robot) for each type of advertising message | Not supported |
| H4 | Significant differences in participants' *purchase intentions* for each type of advertising message | Partially supported for Guilt Appeal |
| H5 | Significant differences in participants' *brand selection* for each type of advertising message | Not Supported |

Over-solicited consumers [104] often disregard marketers' persuasion attempts [105]. It is generally acknowledged in advertising literature that individuals with higher skepticism towards a promotional message would exhibit fewer positive responses towards the product. Moreover, researchers agree that customers who view marketers' messages as carrying a "manipulative intent" are less likely to purchase the product [78].

In this study, the results provide an interesting insight into how a social robot will perform when it is delivering an advertising message. The research has two key theoretical contributions, (1) how effectively can a social robot deliver an emotional message and (2) how effectively can a social robot change consumer's decisions. This is the first study to provide an empirical result to answer these questions to reduce the consumption of high-sugar energy drinks.

Participants who were exposed to the guilt message identified that the robot's manipulative intent was higher than the humour message. This indicates that the robot's verbal and non-verbal cues are not effective at delivering a guilt message. For example, the built-in features for Nao's verbal and non-verbal guilt cues are limited. The social robot has a limited range of voice, tone, facial expression, and eye movement and these are seemingly not sufficient at delivering a guilt message. The reduced ability of the Nao robot to depict facial expressions is acknowledged [106]. In this study, Nao's body language and tone were contributing factors in the poor delivery of the guilt message. However, compared that to a humour message, the robot was able to make participants' laugh with its joke-based dialogue. The use of "hahaha voice over", laughs, hands over the mouth, and giggling to deliver the humour message had a profound effect on consumers' responses. This suggests that social robots such as Nao can display positive emotions effectively but cannot display negative emotions (e.g., guilt) effectively. We invite the reader to consult with [107] for an overview of the importance of how the range of non-verbal robot gestures can influence humans.

Consumers' perceived manipulative intent is shaped by the perceived credibility of the message [60]. Therefore, it is quite possible that a straightforward message or a guilt appeal message is not consistent with customers' expectations from a robot [60]. The Expectancy

Disconfirmation Theory supports these findings and suggests if the message is not in line with what the person believes in, they will question the message and feel that the robot is trying to manipulate them [108]. For example, a humourous message delivered by a social robot is in line with customers' expectations of a social robot, e.g., the robot should be entertaining and fun. Thus, the participants are more likely to accept the humour message than the guilt message. This has also resulted in higher intentions to purchase the product for the humour message condition.

## 6. Limitations and Future Research Directions

Due to the exploratory nature of the research, it has a few limitations to consider. Firstly, the data was collected with undergraduate students. Therefore, future studies should explore the use of social robots with other demographics. The study is also limited to examining the effectiveness of anticipatory guilt in one context only (energy drinks). Future studies should examine other types of guilt messages (reactive and existential) and in a durable or service context. Past research indicates that guilt performs differently in durable, non-durable, and services contexts [56]. The study was also conducted as a one-off interaction, where persuasion or behaviours change in the field of HRI is now being analysed in a longitudinal fashion [109]. Further, the study used only one type of social robot (Nao). An even more anthropomorphic social robot might perform very differently. Finally, this was tested in a lab environment, many factors such as mood, lighting, sound, and visuals were controlled. Future studies should replicate the study in the real world to provide more ecological data. In the future, it would be interesting to conduct a comparative study between social robots and human agents using the same appeals, same message, and in the same business environment.

**Author Contributions:** Conceptualization, O.M. and M.L.; Methodology, O.M. and M.L.; Formal analysis, I.K. and A.K.; Data curation, O.M. and S.S.; Writing—original draft preparation, I.K., A.K., O.M. and M.L.; Writing—review and editing, I.K., A.K., O.M. and M.L.; All authors have read and agreed to the published version of the manuscript.

**Funding:** This research received no external funding.

**Institutional Review Board Statement:** This study has been approved by the Human Research Ethics Committee at Western Sydney University. The ethics reference number is: H13082. Informed consent was obtained from all subjects involved in the study.

**Informed Consent Statement:** Informed consent was obtained from all subjects involved in the study.

**Data Availability Statement:** Data available on request due to ethical obligations. The data presented in this study are available on request from the corresponding author.

**Conflicts of Interest:** The authors declare no conflict of interest.

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
