# Peer review of "The Effectiveness of Robot-Enacted Messages to Reduce the Consumption of High-Sugar Energy Drinks"

_informatics, doi:10.3390/informatics9020049_

Round 1

Reviewer 1 Report

Abstract

            The abstract of this research paper does a succinct job of pointing out the main purpose of the study which is to look at how robot generated messages can influence people’s intake of high sugar energy drinks. The exploratory nature of the study is something that has been pointed out in the abstract, and the fact that an experimental research design has been made upon to hit upon the results and the conclusion of the study is made known in the abstract as well. The abstract serves its purpose of providing an overview of the main theme and concept of the study, the methods used to conduct the study as well as the results or outcome of the same.

Literature Review

            It is clear from reading the literature that has been analyzed that the researchers have engaged in the reading of a wide range of texts on the subject of robotics, and the role that robotics can play in the domain of marketing. There is a plethora of literature that has been analyzed by the authors of this piece on the subject of social robotics and how it is increasingly being used in the domain of marketing, and the type of message appeals that robots’ are seen to generate is something that has been discussed here as well. The gaps in the literature are adequately pointed out, and efforts are made to ensure that the rationale or the purpose of the study is something that has been expressed with care and clarity.

Methodology

            The methodology for study is a matter that has been well discussed and there is an elaborate description that has been provided about the sample population for the study and the experimental groups and overall approach that was used to conduct and conclude the study in a systematic way. The study was conducted in the context of students who were enrolled in the University of Sydney, with all experimental rules as well as protocols being followed at the time of the study being undertaken. Humor appeal, non-emotional appeal and guilt appeal which are the three forms of appeals that are typically associated with robotics, were judged in the course of performing this study.

Results

            The results of this study have been presented in quite a systematic way, with plenty of charts, graphs and diagrams having been used at the time of presenting the results of the study. The authors of the piece have managed to establish the fact that there is a direct connection between the messages that are generated by social robots and the intake of high energy drinks by the participant population, with most of the participants being guided or influenced by the emotional appeal or value of such messages.

Conclusion

            The conclusion of the research project is something that has been presented in quite a constructive way, and the fact that there is scope for future research to be carried out on this subject, is a fact that has been made known quite well, with reference also being made to the funding and other important aspects associated with the current study.

Author Response

Thank you so much for your reviews

Reviewer 2 Report

Authors are proposing an interesting topic of research. In order to have a proper approach from a methodological point of view, I advise the authors to take account on the following comments:

  • I advise the authors to extend the part referring to the relevant literature in the field with several concrete examples of other research as much similar as they can find with the one proposed within this paper (or at least other examples of research that are able to describe in a clear manner the way in which robots are already used in a commercial environment, interacting with people and deliver various messages to them)
  • The part of the text before the hypothesis (especially from line 155 – 242) has to be rewrite and re-organized in order to explain and to present very clear the motivation behind every advanced hypothesis. Authors will present arguments and other results that can sustain in a clear way the positive statement of every hypothesis
  • At line 376, authors are presenting an image entitled setup layout. The idea to present actual, real images from within the experiment is very good but, in my opinion maybe is better to have another image with a more clear angle regarding the robot, and the room already prepared to meet the participants (cosy, without any other elements that can distract like open boxes etc). If the experiment protocol permit it (regarding the confidentiality aspects) is best to use an image from during the experiment itself with one of the participants (with the camera focused on the robot, with an angle that not disclose the face of the respondent eventually..). This can give a more proper feeling about the experiment for any future reader of the paper.
  • On the basis of which values of Sig from tables 7 and 8, respectively, do you consider that the results have been validated?
  • Table 9 states that hypothesis 5 is supported. On what basis?
  • Below, in the explanations, a table corresponding to the bivariate analysis is indicated and it is stated that the application of the Chi square test does not statistically validate the differences between the groups (i.e hypothesis 5). So in the end what is the status of hypothesis 5? However, the authors launch some explanations stating that "advertisers can use any type of advertising appeal as a preventive strategy to reduce the consumption of high sugar energy drink" - a general statement that cannot be verified in any context of consumption. Below, the appeal to the Hierarchy of Effects model is vague, to support another "extremely" general statement - "consumer behavior does not change instantly"… "there are several stages between ignorance of a product / brand and the actual purchase of the product". I recommend that the authors use more validated or statistically invalidated results and comment on them (for example, to elaborate on the invalidation of hypothesis 5 - it may be necessary to resume the experiment on several subjects, etc.). In order to meet the requirements related to a quality article, more attention is needed on these requirements related to the validation / invalidation of advanced hypotheses.
  • The references has to be mandatory rewritten! Every Journal ask for a specific set of requirements regarding the form and the way in which the reference list should be written (see the authors guide on the website). Authors are strongly advised to reconsider entirely all the lines regarding references because the lack of homogeneity and different information that is present for different sources. Order of Initials of authors is wrong, names of publishing houses cities etc. are missing, etc. Another element that has to be taken into consideration is referring to the need to update some of the sources cited – for example: …”Kotler, P., et al., Principles of marketing practice. Hall International Editions, Englewood Cliffs, 1994”…given as a reference for a very general idea like:…”An emotional appeal consists of positive (e.g. humor) or negative (e.g. guilt) emotional elements”…The well-known author has many other very new sources (papers, books) from where authors can find the same idea or something similar in order to quote.

The above observations and comments are made solely in order to help authors to raise properly the quality of the paper for publication purposes. 

I wish succes for the authors in their efforts!

Author Response

Please see the attached responses

Reviewer 3 Report

The topic is  very interesting. The paper is overall readable. In my opinion there are some points to improve.  The leterature section is clear and well structure.    However, I would recommend doing a bibliographic study on the reference scientific platforms such as scopus or web of science to strengthen the bibliography. Despite being a bibliography with a good scientific solidity, I noticed that just over 40% of the references refer to the last ten years and just over 20% of the last five years. 

In the section "3 Methodology"in reference to lines 257-259, you should be able, if it were possible to identify any study that has adopted a procedure similar to the one you have used.

You fill in the "Author Contribution" 

Author Response

Thank you so much for your reviews. We have attached the responses here. 

Reviewer 4 Report

Dear Author/s,

Your work entitled "The Effectiveness of Robot-Enacted Messages to Reduce the Consumption of High-sugar Energy Drinks" is interesting and the majority of the work is clear and well-explained.

The introduction and the literature section provide the information due to understanding the objectives of the study. Findings are coherent with the declared scope and, in my humble opinion, are satisfactory and well-explained. Moreover, results are compared with coherent literature to amplify the positive value of the evidence. Lastly, the “conclusion” reflects the structure of the paper and some indications about limitations and feasible future research are provided.

However, in my opinion, the present form of the paper needs some integrations in the Methodology section before considering it eligible for publication in the Informatics journal.

In Particular, I suggest integrating some indication as to the definition of the sample that should help to understand more in-depth the clear limitations declared. Furthermore, I think that this section should be supported by other and consistent literature to improve the scientific soundness of the paper.

Therefore, in the present form, the paper could be eligible for the publication minor revision and as soon as other reviewers and Editors will provide a positive evaluation.

Best regards.

Author Response

Thank you so much for your reviews. Please see the attached responses. 

Reviewer 5 Report

This exploratory study examines the effectiveness of social robots’ ability to deliver advertising messages using different ‘appeals’ in a business environment. It explores the use of three types of message appeals in a human-robot interaction scenario: guilt, humour and nonemotional. I offer a few suggestions for modification.

Originality:  
Does the paper contain new and significant information adequate to justify publication?

There have been many concrete empirical studies on humour or guilt, so the contribution of this study is relatively weak.

Methodology:  
Is the paper's argument built on an appropriate base of theory, concepts, or other ideas?  Has the research or equivalent intellectual work on which the paper is based been well designed?  Are the methods employed appropriate?

Table 5. "The robot tried to manipulate the audience in ways I do not like", "I was annoyed by this robot because it was trying to inappropriately man age or control the audience" in Measurements: Reliability of Survey Items.
Can the author add that the above questions are reverse coded?
And is it because of reverse coded, resulting in lower Scale Reliability (0.610-0.720)?

Table 8. Humour-Guilt (.3824*), Humour-Guilt (.7557*) in Post Hoc Tests. The author can add an explanation of the reasons for achieving positive salience.

Implications for research, practice and/or society:  
Does the paper identify clearly any implications for research, practice and/or society?  Does the paper bridge the gap between theory and practice? 

Although the title is "The Effectiveness of Robot", the study design does not convey the difference in effectiveness achieved by the robot. For example, at the same time, the control group is also based on the notification from family members, or from the reminder from the notice board, so that the effectiveness of the robot notification can be more clearly verified.

The results of the paper mention (1) how effectively can a social robot deliver an emotional message and (2) how effectively can a social robot change consumer’s decisions.
But if it is verified from the research results, is it because of the influence of the robot? Or is it because of the effect of humour or guilt information? Even if the experimental tool is changed into a communication tool other than the robot, is it possible to cause the same communication effect and research? result? 
So is it because of the influence of the message? Or the influence of the robot? Both of these points may cause readers to doubt the results of this study.

Author Response

Thank you so much for your review. Please see the attached responses. 

Round 2

Reviewer 2 Report

I noticed that the authors took into consideration most of the observations and understood their significance.

In the following I insist on three issues which, in my opinion, still need to be corrected accordingly:

1.        Your statement according that, the table no 8….”provide a validated finding between the three groups (Control, Guilt and Humor) for each of the hypotheses” how is sustained by the values from the table?

I strongly recommend for a more clear text to introduce a reference to a well-known bibliographic source of a statistical nature, referring to Sig's values for the Post Hoc Test to support the validation of your hypotheses!

2.        At line 578, I recommend to change the expression……”we invite the reader to consult” with …..we can take into consideration the work of………and……..or completed with…………………!!

3.        Bibliographic references must be corrected again!

Below are two examples from your properly corrected bibliography that you have as a model. The rest of the bibliography works must be arranged in a similar way!!

Ahearne, M.; Jelinek, R.; Jones, E. Examining the effect of salesperson service behavior in a competitive context. Acad. Mark. Sci., 2007, 35, 603-616. https://doi.org/10.1007/s11747-006-0013-1

Bacharach, S.B.; Bamberger, P.; Conley, S. Work‐home conflict among nurses and engineers: Mediating the impact of role stress on burnout and satisfaction at work. J. Organ. Behav., 1991, 12, 39-53. https://doi.org/10.1002/job.4030120104

Success!

Author Response

Thank you for reviewing. Attached are the responses. 

Reviewer 5 Report

The author has revised the paper according to the reviewer's suggestion.

Author Response

Thank you so much for reviewing.